# Influence of Nitrogen Supply on Growth, Antioxidant Capacity and Cadmium Absorption of Kenaf (*Hibiscus cannabinus* L.) Seedlings

**DOI:** 10.3390/plants12234067

**Published:** 2023-12-04

**Authors:** Wenlue Li, Changli Chen, Yong Deng, Xiahong Luo, Tingting Liu, Xia An, Lina Zou, Mingbao Luan, Defang Li

**Affiliations:** 1Institute of Bast Fiber Crops, Chinese Academy of Agricultural Sciences, Changsha 410205, China; kobe1924@163.com (W.L.); dengyong@caas.cn (Y.D.); luanmingbao@caas.cn (M.L.); 2College of Plant Science & Technology, Huazhong Agricultural University, Wuhan 430070, China; 3Zhejiang Xiaoshan Institute of Cotton & Bast Fiber Crops, Zhejiang Institute of Landscape Plants and Flowers, Zhejiang Academy of Agricultural Sciences, Hangzhou 311251, China; chenchangli@zaas.ac.cn (C.C.); luoxh@zaas.ac.cn (X.L.); liutt@zaas.ac.cn (T.L.); anxia@zaas.ac.cn (X.A.); zoulina@zaas.ac.cn (L.Z.); 4National Nanfan Research Institute (Sanya), Chinese Academy of Agricultural Sciences, Sanya 572024, China

**Keywords:** kenaf, cadmium, absorption, antioxidant capacity, translocation factors

## Abstract

Kenaf (*Hibiscus cannabinus* L.) is considered suitable for the remediation of cadmium (Cd)-contaminated farmlands, because of its large biomass and resistance to Cd stress. The addition of nitrogen (N) fertilizer is an important measure used to increase crop yields, and it may also affect Cd accumulation in plants. To clarify the effects of different forms and concentrations of N on plant growth and Cd absorption in kenaf, a hydroponic experiment was conducted using three N forms (NH_4_^+^–N, NO_3_^−^–N and urea–N) at four concentrations (0, 2, 4 and 8 mM, 0 mM as control) under Cd stress (30 μM). The plant growth, the antioxidant enzyme activity and the Cd contents of various parts of the kenaf seedlings were measured. The results showed that the N form had the greatest impact on the growth of the kenaf and the absorption and transport of the Cd, followed by the interaction effect between the N type and the concentration. Compared to the control, the addition of N fertilizer promoted the growth of kenaf to varying degrees. Among all the treatments, the use of 2 mM of NO_3_^−^–N enhanced the biomass and Cd accumulation to the greatest extent compared to CK from 2.02 g to 4.35 g and 341.30 μg to 809.22 μg per plant, respectively. The NH_4_^+^–N significantly reduced the Cd contents of different parts but enhanced the translocation factors of Cd stem to root (TF S/R) and leaf to stem (TF L/S) by 34.29~78.57% and 45.10~72.55%, respectively. The peroxidase (POD), superoxide dismutase (SOD) and catalase (CAT) enzyme activities of the kenaf increased with the N treatments, especially with NH_4_^+^–N. Overall, applying low concentrations of NO_3_^−^–N can better promote the extraction of Cd by kenaf.

## 1. Introduction

Cadmium (Cd) is a highly toxic and non-essential heavy metal that is difficult to degrade. Although Cd is not abundant in soil, it is easily absorbed by plants and impairs their growth. It enters the human body through the food chain, where it causes various diseases [1,2,3]. China has a large population but limited arable land resources, and approximately 1/6 of the land is polluted with Cd to varying degrees [4,5]. The remediation and safe use of Cd-contaminated farmlands are important to ensure the food supply, and considerable research on the restoration and management of Cd-contaminated farmlands has been conducted in recent years [2,4,5,6,7]. For farmlands with mild-to-moderate heavy metal pollution, the main measures employed are to rescue Cd bioavailability in the soil by using low-uptake varieties and adopting agronomic measures to reduce the Cd contents of grains. For heavily polluted farmlands, economical plants that do not enter the food chain are used as substitutes for planting, and to restore the basic ecological functions of polluted farmland while generating economic benefits [5,6,7].

*Hibiscus cannabinus* L., also known as kenaf, is an annual bast fiber crop and an industrial crop used in a wide range of applications (such as textile production, papermaking, construction materials, hemp plastic and activated carbon) [8,9,10,11]. Related research has indicated that kenaf can adapt to high concentrations of Cd stress (4.9 mg·kg^−1^) and extract a certain amount of Cd (74.42~149.17 g·hm^−2^) [9,12]. In addition, the heavy metal content extracted from bast fiber and stem xylems is known to meet the diamond standard for ecological textiles (<40 mg/kg) and the limits for indoor decoration and the refurbishing of materials (<75 mg·kg^−1^) [12,13]. Therefore, the use of kenaf is widely considered in the reclamation of heavy-metal-contaminated farmlands [9,11,12].

The application of nitrogen (N) fertilizers has become an essential approach for increasing crop yields in China [14,15,16,17], even in heavy-metal-contaminated soils. Research has shown that the application of N fertilizer can alleviate the damage caused by Cd stress in plants and affect the absorption and accumulation of Cd in plants [1,18,19]. Studies have suggested that ammonium N fertilizer can lead to soil acidification and increased plant Cd contents [20,21]. However, other studies have found a synergistic relationship between the absorption of NO_3_^−^ and Cd^2+^ by plants; and compared to ammonium N fertilizer, nitrate N fertilizer has been found to enhance the absorption of Cd by plants [22,23]. In addition to the fertilizer form, the N concentration can also affect Cd absorption by plants [24,25], but research results have varied greatly because of the differences in the experimental materials and methods used [23,25,26]. In summary, although the effects of N fertilizer on different plants vary, the use of fertilizers to regulate the uptake of heavy metals by plants is considered the most economical and least disruptive method for plant growth [1,19].

Related research has not been conducted using kenaf. Therefore, kenaf (Qingpi no. 3) was used as the experiment material in this study. It was grown in a hydroponic system under Cd (30 μM) stress, where it was supplied with three N forms (NH_4_^+^–N, NO_3_^−^–N and urea–N) at four concentrations (0, 2, 4 and 8 mM, and 0 mM as the control). To screen out the applicable N form and concentration that enhanced Cd accumulation in kenaf, the phenotypic traits of the kenaf seedlings and the physiological changes occurring in relation to the Cd stress and the Cd content were investigated, in addition to accumulation (total Cd in plant) and the translocation factor TF (the ratio of the metal concentration in the stems or leaves to that in the roots) were investigated [9,13], The findings in this study provide a theoretical basis for the remediation of Cd-contaminated farmlands using kenaf.

## 2. Results

### 2.1. Phenotypic Traits

After four weeks of treatment, significant differences in kenaf growth were observed under Cd-stress conditions (Table 1). The two-way ANOVA results (Table 2) indicated that the N form and the interaction between the N form and concentration significantly affected multiple agronomic traits, except for the stems’ dry weight.

The plant heights ranged from 34.08 cm to 49.92 cm, the stem diameters ranged from 2.70 mm to 3.84 mm, the maximum root lengths ranged from 17.42 cm to 29.83 cm, and the dry weights of the roots, stems and leaves ranged from 0.49 to 0.81, 1.09 to 1.99, and 0.31 to 1.59 g·plant^−1^, respectively. Compared with the N-deficiency (CK), the application of N promoted the growth of kenaf to varying degrees (except for the root dry weight). Among all the agronomic traits, the increase in the leaf dry weight was the most significant, ranging from 148.38% to 412.90%, followed by stem dry weight, which increased by 22.02% to 82.57% (except for the group with 8 mM NO_3_^−^–N). Within the experimental concentration range, the growth of kenaf in the urea- and NO_3_^−^–N groups showed an increasing trend, followed by a decreasing trend with increasing N concentration. The best growth performance at 4 mN was seen with urea–N, whereas NO_3_^−^–N provided the best growth performance at 2 mM. The growth potential of the kenaf in the presence of NH_4_^+^–N gradually increased with the increasing N concentration.

### 2.2. Chlorophyll Content

The chlorophyll content is an important indicator of plants’ responses to stress. The two-way ANOVA results revealed that the N form and the interaction between the N form and the concentration significantly affected the chlorophyll a and chlorophyll b contents, whereas the N concentration only significantly affected the chlorophyll b content (Table 2). The chlorophyll a, chlorophyll b and total chlorophyll contents of the plants in each treatment ranged from 0.57 to 0.90 mg·g^−1^ FW, 0.29 to 0.70 mg·g^−1^ FW and 0.87 to 1.62 mg·g^−1^ FW, respectively (Figure 1). Overall, the chlorophyll contents of the plants in the NO_3_^−^- and NH_4_^+^–N treatments were higher than those in the urea–N treatments, with the highest total chlorophyll content observed in the 4 mM NO_3_^−^–N and 8 mM NH_4_^+^–N treatments, which increased by 60.20% and 65.31%, respectively, compared to the CK.

### 2.3. Uptake, Transfer and Accumulation of Cd in Different Parts of Kenaf

#### 2.3.1. Cd Content

The Cd contents of kenaf leaves, roots and stems are shown in Figure 2, where significant differences between the treatments are evident. The two-way ANOVA results (Table 2) indicated that the N form and concentration, as well as their interaction, had significant effects on the Cd contents of different parts of the kenaf. The variance of the N form was greater than that of the concentration and the interaction effect of the N form and concentration (Table 2), indicating that the N form had a greater effect on the Cd absorption than the N concentration.

The root system was the major organ accumulating Cd in the kenaf, and the Cd contents ranged from 187.50 mg·kg^−1^ to 1521.50 mg·kg^−1^ (Figure 2). At equal N concentrations, the root Cd concentration was the highest under the NO_3_^−^–N treatment, followed by the urea–N treatment, and it was the lowest under the NH_4_^+^–N treatment. Compared with the CK, the NO_3_^−^–N treatment significantly increased the Cd content by 104.44~122.97%, while the NH_4_^+^–N reduced the content by 57.78~72.52%. For the Cd contents of the stems, there was no significant difference between the Urea- and NO_3_^−^N treatments, which ranged from 444.67 mg·kg^−1^ to 523 mg·kg^−1^, but they were significantly higher than that of the NH_4_^+^–N treatment (212.58~290.68 mg·kg^−1^). The Cd contents of the leaves ranged from 164.92 mg·kg^−1^ to 301.39 mg·kg^−1^. With an increase in the N concentration, there was a gradual increase in the Cd contents of all the kenaf parts with urea–N, while the Cd content of the root system treated with NO_3_^−^–N and the Cd contents of all the kenaf parts treated with NH_4_^+^–N showed an initial increase and a final decrease.

#### 2.3.2. Translocation Factor

The corresponding TFs were calculated based on the Cd contents of different plant parts (Figure 3). The two-way ANOVA results (Table 2) showed that the N form and concentration, as well as the interaction between hem, significantly affected the shoot/root (TF _S/R_) and leaf/root (TF _L/R_) TFs, among which the effect of the N form was the strongest. Generally, the TF _S/R_ of different treatments ranged from 0.30 to 1.25, in the following order: NH_4_^+^–N > CK > Urea-N> NO_3_^−^–N. Specifically, compared with the CK, the TF _S/R_ under the NH_4_^+^–N treatments increased by 34.29~78.57%, while those under the NO_3_^−^–N treatments decreased by 49.98~57.14%. The TF L/R values for the different treatments ranged from 0.16 to 1.09, and the highest values were also observed in the NH_4_^+^–N treatments, which increased by 102.27~203.62% compared to the CK. With an increase in the N concentration, the TF _S/R_ and TF _L/R_ under different treatments showed different tendencies. Specifically, the TF _S/R_ and TF _L/R_ under the NH_4_^+^–N treatments first increased and then decreased, whereas those of the urea–N treatments showed a linear decrease, and no significant changes were observed for those under the NO_3_^−^–N treatment. In general, NH_4_^+^–N was more beneficial for the transport of Cd to the aerial parts, whereas NO_3_^−^–N decreased the transport of Cd to the stem.

#### 2.3.3. Cd Accumulation in Different Parts of Kenaf

The total amount of Cd absorbed by kenaf is an important indicator of remediation effectiveness. The amount of Cd accumulated by each plant part was calculated by combining the biomass and the Cd content of the kenaf (Figure 4). The two-way ANOVA analysis (Table 2) revealed that the N form and the interaction effect between the N form and the concentration significantly affected the amount of Cd accumulated in various parts of the kenaf, and the N form had the greatest effect. The results in Figure 4 show that the urea- and NO_3_^−^–N treatments significantly increased the amount of Cd accumulated in the whole plant, with the highest amounts accumulated under the 2 mM and 4 mM NO_3_^−^–N treatments, which increased by 137.10% and 96.47%, respectively, compared to the CK treatment (Figure 4d). Specifically, the amount in the roots increased by 165.81% and 112.15% (Figure 4a, that in the stems increased by 70.26% and 43.34% (Figure 4b), and that in the leaves increased by 434.46% and 373.27% (Figure 4c), respectively. With the increasing N concentrations, different N treatments showed different Cd absorption trends across the whole plant. The total amount of Cd in kenaf under the urea–N treatment showed a gradually increasing trend, and that under the NO_3_^−^–N showed a gradually increasing trend followed by a decreasing trend, whereas that under the NH_4_^+^–N treatment showed a decreasing trend followed by an increasing trend.

### 2.4. Antioxidant Enzyme Activity and Proline Content

After applying different N fertilizer treatments for 7 days under Cd-stress conditions, the activities of SOD, POD and CAT, as well as the MDA contents of the roots, were tested, and the results are shown in Table 3. Compared to the CK, the addition of N treatments significantly increased the activities of SOD, POD and CAT, with the greatest increase observed in the NH_4_^+^–N treatment, and SOD, POD and CAT activities increased by 112.51~177.35%, 149.18~180.21% and 240.14~328.75%, respectively. The MDA content of each treatment ranged from 4.54 to 6.34 nmol·g^−1^ FW. The MDA content in the urea- and NO_3_^−^–N treatments increased with increasing N concentration, whereas that in the NH_4_^+^–N treatments showed a trend of first increasing and then decreasing.

## 3. Discussion

### 3.1. Under Cd Stress, the Supply of N Promoted the Growth of the Aerial Part

Nitrogen is an essential element for plant growth, and it plays an important role in cellular genetics and metabolism [1], as well as in crop yields [15,16,17,19]. Studies have shown that increasing N fertilizer application can alleviate plant toxicity symptoms in Cd-contaminated soils [27,28], and with an increase in the amount of N fertilizer added, the plant biomass gradually increases under Cd stress [27,28,29,30]. Different types of N fertilizer have different effects on plant growth under Cd stress. For example, NH_4_^+^–N fertilization was shown to produce better growth in corn and rice [31,32], and the yield of rice treated with urea–N fertilizer was found to be higher than that treated with other N fertilizers [33].

In the present study, the effects of different forms and concentrations of N fertilizer on the biomass, agronomic traits and chlorophyll contents of kenaf were compared. The results illustrated that compared with N-deficiency treatment (CK), the supply of N can promote the growth of kenaf stems and leaves (Table 1), which is consistent with previous studies [31,34]. The two-way ANOVA analysis indicated that the N form had the greatest effect on the growth of the kenaf, followed by the interaction effect of the N form and concentration (Table 2). Comparing the biomass of the kenaf under different N treatments, the treatment with NO_3_^−^–N ranked first (Table 1), which is consistent with the N fertilizer study on brassica [35], but opposite to the results obtained using rice [22,32], indicating that kenaf has nitrification characteristics and that NO_3_^−^–N has a more significant effect on promoting its growth [19]. Different N forms significantly change chlorophyll contents in plants under Cd stress [32,36]; urea–N shows a great advantage in improving chlorophyll content [36]. In this study, NO_3_^−^-and NH_4_^+^–N enhanced the chlorophyll content more significantly than the CK and urea–N treatments (Figure 1), further illustrating the differences in fertilizer effectiveness between rice and kenaf.

### 3.2. NO_3_^−^–N Promotes the Absorption of Cd by Kenaf, and the Total Accumulation of Cd Was Highest under Low-Concentration Conditions

A considerable amount of research has been conducted on Cd absorption in relation to the N fertilizer form or concentration [37,38,39,40]. Some studies have revealed that the Cd content is significantly positively correlated with the amount of N fertilizer applied [37,38], while another study arrived at the opposite conclusion [39], and a further study showed that Cd accumulation in plants is not linearly related to the amount of N fertilizer applied. In addition, one study found that under a moderate urea treatment, the Cd content and total accumulation in corn were higher than the values obtained under other N-concentration treatments [30]. In terms of studies on the N form, several have shown that the addition of NH_4_^+^–N fertilizer can lower soil pH and increase the available Cd content in the soil, which thus increases the plant Cd content and accumulation [21,26,33]. Hydroponic experiments have shown that NO_3_^−^–N fertilizer is beneficial for absorbing more Cd [21,22,35,40], and a further analysis revealed that NO_3_^−^ and Cd exhibited synergistic effects, whereas NH_4_^+^ and Cd exhibited antagonistic effects [32,41].

In this study, we compared the Cd contents of kenaf leaves, stems and roots under different N conditions. The total amount accumulated in each part was calculated in combination with the biomass. The results showed that the N form and the interaction effect between the N form and the concentration significantly affected the content and accumulation of Cd in different parts of the kenaf (Table 2). The Cd contents in the leaves, stems and roots ranged from 164.92 to 301.39 mg·kg^−1^, 212.58 to 523.91 mg·kg^−1^ and 187.50 to 1521.50 mg·kg^−1^, respectively, which was consistent with previous hydroponic experiments on kenaf under Cd stress [10,42,43]. The difference in the Cd content of the root system of the kenaf was the most significant under different N forms, specifically NO_3_^−^–N > urea–N > NH_4_^+^–N (Figure 2). Compared to the CK, the NO_3_^−^- and urea–N treatments significantly increased the Cd in the roots. In contrast, the NH_4_^+^–N treatments significantly reduced the Cd contents of various plant parts (Figure 2); this result is consistent with those of previous studies [29,33,44]. Progress has been made in explaining the mechanism associated with the N form and plant Cd absorption. First, NO_3_^−^ significantly increases nitrate reductase (NR) activity, whereas NH_4_^+^ significantly inhibits NR activity and ultimately affects the level of nitric oxide (NO) [45,46]. Furthermore, NO can increase the contents of pectin and hemicellulose, fix Cd in the roots and reduce its transport to the aerial parts [47], which may also explain the difference between the translocation factors of the NO_3_^−^ and NH_4_^+^–N treatments. Second, NO_3_^−^–N can affect the absorption of Cd by regulating Fe transporters. According to a previous study on rice, NO_3_^−^–N regulated the expression of *OsIRT1* and *OsNramp1* genes in rice to increase the absorption of Cd in rice roots [25]. In addition, when the ratio of NO_3_^−^/NH_4_^+^ in the nutrient solution was increased, the expression of the *OsIRT1*, *OsNramp5* and *OsHMA2* genes in roots increased, leading to an increase in the Cd content [48]. Finally, studies have shown that NO_3_^−^ transporters significantly affect Cd absorption [41,49].

This study also discovered that, with urea–N, the Cd contents in various parts of kenaf gradually increased with increasing concentrations (Figure 4), which agrees with the previous results [30]. The concentration of Cd in the roots treated with NO_3_^−^–N showed a trend of first increasing and then decreasing with an increase in the N concentration (Figure 4), which is similar to the results of a previous study, which concluded that the highest accumulation of Cd occurs in plants treated with low concentrations of NO_3_^−^–N fertilizer [50]. The concentration of Cd in various parts of the plant treated with NH_4_^+^–N showed a trend of first decreasing and then increasing with an increase in the N concentration (Figure 4), which is consistent with a study conducted on *Kandelia obovata*, which found that high concentrations of NH_4_^+^ promoted Cd accumulated in roots [51]. The comparison of the total amount of Cd accumulated in kenaf under different N treatments showed that the 2 mM NO_3_^−^–N treatment ranked first (20.68~259.82% higher than that of other treatments) (Figure 4), which indicated that adding a certain amount of NO_3_^−^–N can enhance Cd extraction by kenaf.

### 3.3. NH_4_^+^–N Promotes the Transfer of Cd in Kenaf to the Aerial Parts

Following Cd’s adsorption in the roots, a large proportion is trapped in the roots and stored in vacuoles [52], while another part complexes with amino acids and citric acid. It is transported to the aerial parts of the plant (through loading, transportation and unloading in the xylem under the combined function of transpiration and root pressure) and then allocated to the vegetative organs, such as the leaves [3]. Different forms of N significantly affected the transport of Cd to aerial parts, and one study found that NH_4_^+^–N promoted the transfer of Cd from straw to grain and the roots to the aerial parts [20,33].

In the present study, the form and concentration of N and their interactions significantly influenced the transfer of Cd to the aerial parts (Table 2). Compared with the N-deficient treatment, the addition of NH_4_^+^–N significantly increased the TF _S/R_ and TF _L/R_, whereas the NO_3_^−^–N significantly reduced the TF _S/R_ and TF _L/R_ (Figure 3), which is consistent with previous research [20]. According to a previous study, a ratio of 50/50 NH_4_^+^/NO_3_^−^ is beneficial for plant Cd absorption and transport [53]. We thus consider that it may be possible to apply NH_4_^+^- and NO_3_^−^–N in a certain proportion to increase the Cd contents of both aboveground and underground parts.

### 3.4. Applying N Fertilizer Promotes Antioxidant Enzyme Activity of Kenaf and the NH_4_^+^–N Treatment Showed the Most Significant Enhancement

The MDA is a product of lipid peroxidation, which is considered an indicator of tissue damage. Plants can resist Cd stress by increasing the activity of antioxidant enzymes and antioxidants [31,41,54]. These antioxidant enzymes include SOD, POD, CAT and APX. When plants encounter stress conditions, SOD first converts O_2_^−^ to H_2_O_2_, and then POD, CAT and APX scavenge H_2_O_2_ and other ROS products, such as MDA [42,54]. It has been reported that adding N fertilizer can decrease the MDA and H_2_O_2_ contents under Cd stress [33] and that different N fertilizer treatments significantly affect antioxidant enzyme activity [31,35]. In the present study, the addition of N fertilizer significantly increased the SOD, POD and CAT enzyme activities (Table 3), which is consistent with a previous study [31]. The most signficant antioxidant enzyme activity enhancement was observed under the NH_4_^+^–N treatment, particularly the values of SOD and POD (Table 3), In this respect, it was confirmed that SOD and POD play a crucial role in resisting Cd stress [53]. Unlike in previous studies, the MDA content did not decrease with the addition of the N fertilizer, and this may have been the result of the combined effects of the root Cd content and the antioxidant enzyme activity [41,54].

## 4. Materials and Methods

### 4.1. Plant Material and Experimental Treatments

Qingpi no. 3 was obtained from the Zhejiang Xiaoshan Institute of Cotton and Bast Fiber Crops. Previous research has shown that Qingpi no. 3 has proficient Cd accumulation. Healthy and plump seeds were sterilized with 75% alcohol for 2 min, washed with deionized water three times and germinated on moist filter paper in Petri dishes for 2 d in the dark. Uniform kenaf seedlings were selected and cultured in Hoagland nutrient solution for 7 days. The Hoagland nutrient solution was composed of Na(NO_3_)_2_ 2.0 mM, NH_4_Cl 2.0 mM, CaCl_2_ 2.0 mM, K_2_SO_4_ 0.75 mM, MgSO_4_ 0.5 mM, KH_2_PO_4_ 0.1 mM, MnSO_4_ 0.5 mM, H_3_BO_3_ 10 μM, ZnSO_4_ 1.0 μM, CuSO_4_ 0.20 μM, (NH_4_)_2_MoO_4_ 0.01 μM and Fe-EDTA 100 μM.

Uniformly grown seedlings were then selected and transferred to an improved Hoagland solution in which the N composition was altered. Three N forms (urea, Na(NO_3_)_2_ or NH_4_Cl replacing the mixture of Na(NO_3_)_2_ and NH_4_Cl as the sole N) at 4 N concentrations (0, 2, 4 and 8 mM, 0 mM as control) were employed, and there were 10 treatments in total. Each treatment was set up in triplicate to ensure the reproducibility of the results. Each treatment comprised 4 kenaf seedlings, all treatments were supplied with 30 mM CdCl_2_, and the nutrient solution was refreshed every 5 days. The seedlings were grown in an artificial climate culture room that maintained a day/night cycle of 16/8 h at 25 °C/20 °C, respectively, with relative humidity close to 65% and a light intensity of 12,000 lx.

After 7 days of Cd and different N treatments, 4 plants from each treatment were selected, and the roots and the penultimate functional leaf were collected in liquid nitrogen and stored at −80 °C for further physiological analysis.

### 4.2. Determination of POD, SOD, CAT and MDA Contents

Frozen fresh root samples were obtained to measure the activities of antioxidant superoxide dismutase (SOD), peroxidase (POD), catalase (CAT) and malondialdehyde (MDA), according to a modified method [55,56]. The MDA was determined using the thiobarbituric acid (TBA) test; SOD activity was measured by nitroblue tetrazolium (NBT) test; POD activity was measured using guaiacol colorimetric method; and CAT activity was measured using the molybdate ammonium test. All these indicators were detected using a microplate reader (BioTek Synergy H1, BioTek Instruments, Inc., Winooski, VT, USA). The absorbance was measured at 450 nm for SOD, 470 nm for POD, 405 nm for CAT, 532 nm and 600 nm for MDA. The SOD and POD activities were expressed as enzyme U·g^−1^ (fresh weight, FW), the CAT activities were expressed as enzyme nmol/g/min FW and the MDA contents were calculated as nmol·g^−1^ FW.

### 4.3. Chlorophyll Contents

Frozen leaf samples were collected to measure the chlorophyll contents in accordance with the method presented in a previous study [56], and the supernatant was collected for absorbance at 663 nm and 645 nm using a spectrophotometer (L6, Shanghai Jingke, Shanghai, China). The chlorophyll contents were calculated as mg·g^−1^ FW.

### 4.4. Growth and Biomass Analyses

After 4 weeks of treatment, the plant heights, stem diameters and maximum root lengths of 8 kenaf plants in each treatment were measured. To measure the biomass, the roots, stems and leaves were separated and dried at 105 °C for 30 min and then dried at 80 °C for another 48 h. After cooling, the dry weights of roots, stems and leaves were recorded.

### 4.5. Determining Cd Concentration and Translocation Factor (TF)

To determine the Cd content, dry samples of the roots, stems and leaves (0.2 g) were ground and digested separately in 5 mL of a digestion solution, composed of HNO_3_ and HClO_4_ at a volume ratio of 4:1. After digestion, the samples were diluted to 25 mL with deionized water, and the Cd concentration was determined using an atomic absorption spectroscope and associated method (Z2310, Hitachi, Tokyo, Japan).

The translocation factor was calculated to evaluate the translocation capability of Cd in kenaf tissues [13,56] as follows:

TF _L/R_ = Cd content in leaves/Cd content in roots;

TF _S/R_ = Cd content in stems/Cd content in roots.

### 4.6. Statistical Analysis

All results are presented as the means ± standard (SD). Data taken for the experiments were processed by two-way and one-way analyses of variance and the means were compared using Duncan’s multiple range test at 5% and 1% (*p* < 0.05, 0.01) in SPSS (IBM SPSS Statistic 23.0, IBM SPSS Inc., Almunk, OR, USA). Graphical presentations were performed using using GraphPad Prism (GraphPad Prism 9.0.0, GraphPad Software, San Diego, CA, USA).

## 5. Conclusions

In summary, adding N fertilizer can promote the growth of kenaf to varying degrees, increase POD, SOD, and CAT enzyme activities and affect the concentration and accumulation of Cd in various parts of kenaf under Cd stress. The two-way ANOVA revealed that the N form had the greatest effect on the growth and Cd absorption of the kenaf, indicating that selecting the appropriate N fertilizer type is more important than the amount of N fertilizer used in regulating Cd uptake by plants. The application of N had the most significant effect on the Cd contents of the kenaf roots, with the highest content in the NO_3_^−^–N treatments and the lowest content in the NH_4_^+^–N treatment. Therefore, in production practice, NH_4_^+^ fertilizer can be used to reduce the Cd content and to accelerate the remediation of Cd-contaminated soil by plants. Increases in Cd accumulation can be achieved by applying NO_3_^−^ fertilizer. Among all the treatments, the highest amount of total Cd accumulated in kenaf occurred in the low-concentration NO_3_^−^–N. Compared with the other two N fertilizer types, the NH_4_^+^–N treatment significantly reduced the Cd contents in various parts of the kenaf, whereas it significantly increased the transportation of Cd to the aerial parts.

This study reveals the effects of N fertilizer on kenaf from the perspectives of phenotype, physiological response and Cd accumulation, and thus provides a theoretical reference for the remediation and safe use of Cd-polluted soil. However, crop production is mostly based on the soil environment, and it is difficult to ignore the changes in soil characteristics caused by N fertilizer application, which may indirectly affect the absorption of Cd by plants. Therefore, to better guide production practices, further verification is required.

## Figures and Tables

**Figure 1 plants-12-04067-f001:**
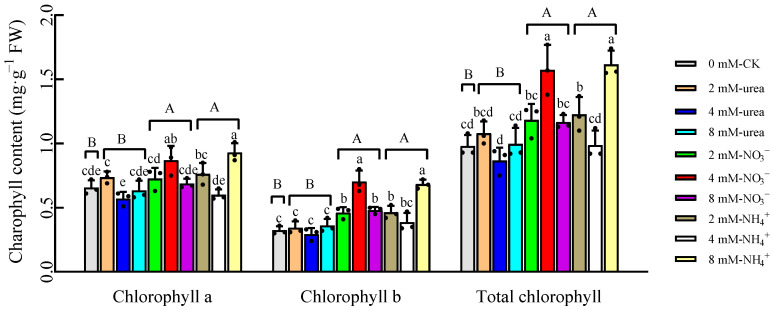
Chlorophyll contents of kenaf leaves under different treatments. Data are means ± SD, the black dots above the bar chart are individual data points, different lowercase letters indicate significant differences between treatments (*p* < 0.05, Duncan) and different uppercase letters represent significant differences between nitrogen fertilizer forms and CK (*p* < 0.05, Duncan).

**Figure 2 plants-12-04067-f002:**
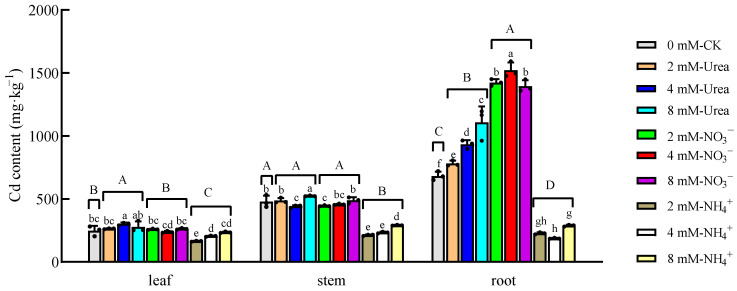
Cd contents of different kenaf parts. Data are means ± SD, the black dots above the bar chart are individual data points, different lowercase letters in each part indicate a significant difference between treatments (*p* < 0.05, Duncan), different uppercase letters in each part indicate a significant difference between nitrogen fertilizer forms and CK (*p* < 0.05, Duncan).

**Figure 3 plants-12-04067-f003:**
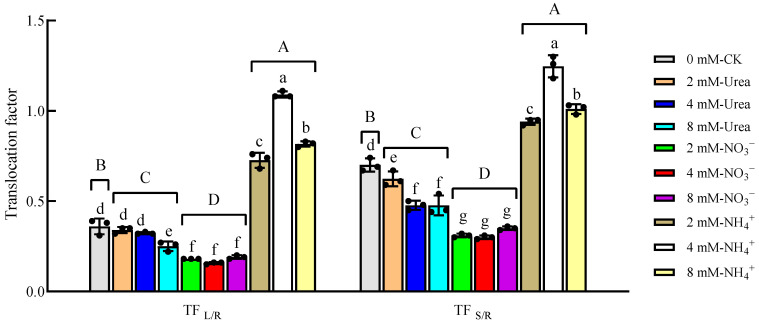
Cd translocation factor of kenaf under different treatments. Data are means ± SD, the black dots above the bar chart are individual data points, different lowercase letters in each part indicate a significant difference between treatments (*p* < 0.05, Duncan), different uppercase letters in each part indicate a significant difference between nitrogen fertilizer forms and CK (*p* < 0.05, Duncan).

**Figure 4 plants-12-04067-f004:**
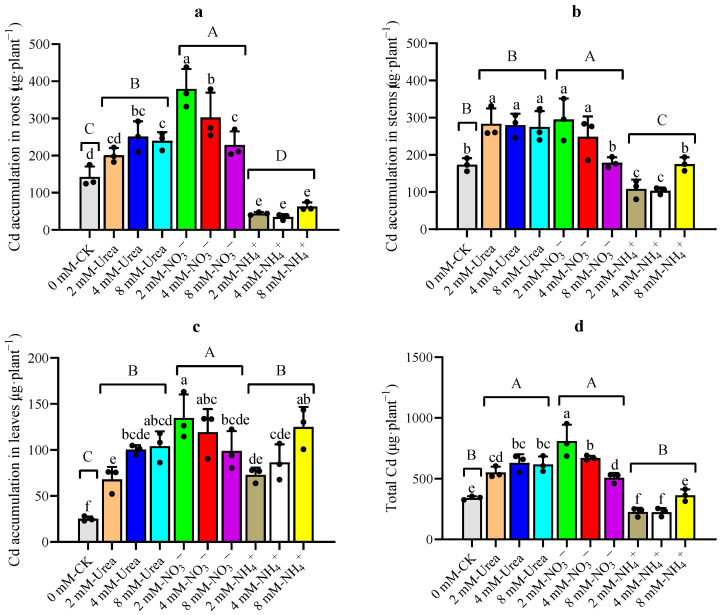
Assessment of Cd uptake under different treatments. (**a**) Cd accumulation in roots, (**b**) Cd accumulation in stems, (**c**) Cd accumulation in leaves, (**d**) the total Cd in kenaf. Data are means ± SD, the black dots above the bar chart are individual data points, different lowercase letters in each part indicate a significant difference between treatments (*p* < 0.05, Duncan), different uppercase letters in each part indicate a significant difference between nitrogen fertilizer forms and CK (*p* < 0.05, Duncan).

**Table 1 plants-12-04067-t001:** Changes in kenaf morphology with different treatments under Cd stress.

Treatment	N Concentration (mM)	Plant Height (cm)		Stem Diameter (mm)		Root Dry Weight (g·Plant^−1^)		Stem Dry Weight (g·Plant^−1^)		Leaf Dry Weight (g·Plant^−1^)		Maximum Root Length (cm)	
CK	0	34.08 ± 3.15 ef	B	3.06 ± 0.23 cd	B	0.62 ± 0.09 abcd	AB	1.09 ± 0.02 c	B	0.31 ± 0.04 e	C	29.83 ± 3.74 a	A
Urea	2	40.54 ± 1.45 cd	A	3.52 ± 0.26 ab	A	0.77 ± 0.10 abc	A	1.57 ± 0.25 ab	A	0.77 ± 0.16 d	B	21.25 ± 3.03 bc	B
	4	46.92 ± 0.80 ab		3.81 ± 0.12 a		0.81 ± 0.12 a		1.90 ± 0.22 a		1.00 ± 0.07 cd		23.83 ± 1.44 b	
	8	45.42 ± 0.29 abc		3.57 ± 0.21 ab		0.66 ± 0.11 abcd		1.75 ± 0.31 ab		1.14 ± 0.23 bcd		25.08 ± 3.17 b	
NO_3_^−^	2	49.92 ± 6.39 a	A	3.68 ± 0.34 a	A	0.80 ± 0.12 ab	AB	1.99 ± 0.36 a	A	1.56 ± 0.27 a	A	24.92 ± 1.04 b	B
	4	43.58 ± 3.39 bc		3.65 ± 0.39 a		0.60 ± 0.15 bcd		1.63 ± 0.35 ab		1.50 ± 0.29 ab		25.42 ± 3.30 b	
	8	38.08 ± 3.09 de		2.70 ± 0.08 d		0.49 ± 0.06 d		1.10 ± 0.08 c		1.13 ± 0.24 bcd		22.42 ± 2.31 b	
NH_4_^+^	2	30.83 ± 1.91 f	B	3.09 ± 0.24 cd	A	0.58 ± 0.03 cd	B	1.33 ± 0.09 bc	A	1.33 ± 0.16 abc	A	16.83 ± 1.01 d	C
	4	32.25 ± 2.84 f		3.21 ± 0.07 bc		0.55 ± 0.12 d		1.53 ± 0.32 abc		1.26 ± 0.26 abc		17.42 ± 0.76 cd	
	8	42.33 ± 1.18 bcd		3.84 ± 0.22 a		0.65 ± 0.10 abcd		1.81 ± 0.20 a		1.59 ± 0.29 a		24.25 ± 1.00 b	

Data in the table are means ± SD, different lowercase letters within a column indicate significant differences between treatments and different uppercase letters represent significant differences between nitrogen forms and CK (*p* < 0.05, Duncan).

**Table 2 plants-12-04067-t002:** The effect of N on agronomic traits, Cd content and Cd accumulation of kenaf under Cd stress based on two-way ANOVA.

Variable	N Form	Concentration	N Form × Concentration
Plant height	27.39 **	0.61	14.33 **
Stem diameter	3.99 *	1.45	12.34 **
Root dry weight	4.96 *	2.8	3.1 *
Stem dry weight	1.57	2.52	5.71 **
Leaf dry weight	11.33 **	0.22	3.63 *
Maximum root length	10.44 **	3.5	4.61 **
Chlorophyll a	7.67 **	2.68	11.65 **
Chlorophyll b	42.08 **	5.96 **	20.49 **
Cd content in leaf	38.9 **	5.21 **	4.53 **
Cd content in stem	441 **	25.8 **	3.42 *
Cd content in root	1292.62 **	12.63 **	13.25 **
Translocation factor leaf/root	2359.46 **	63.25 **	75.72 **
Translocation factor stem/root	1137.41 **	8.34 **	38.91 **
Total Cd in root	127.25 **	1.77	7.11 **
Total Cd in stem	44.64 **	0.87	5.72 **
Total Cd in leaf	6.04 **	2.2	5.79 **
Total Cd in plant	109.37 **	0.68	12.47 **

* and ** denote statistically significant differences at *p* < 0.05 and *p* < 0.01 (Duncan), respectively.

**Table 3 plants-12-04067-t003:** Effects of nitrogen fertilizer on superoxide dismutase (SOD), peroxidase (POD) and catalase (CAT) activities and proline contents in kenaf leaves under Cd stress.

Treatment	N Concentration (mM)	SOD (U·g^−1^ FW)	POD (U·g^−1^ FW)	CAT (nmol·g^−1^ FW)	MDA (nmol·g^−1^ FW)
CK	0	339.36 ± 9.34 h	33.96 ± 1.33 h	9.74 ± 0.50 h	4.54 ± 0.29 e
Urea	2	382.15 ± 13.28 g	63.95 ± 1.07 d	21.82 ± 0.14 g	5.13 ± 0.30 bc
	4	561.36 ± 36.19 e	58.12 ± 0.92 e	25.41 ± 0.82 f	4.74 ± 0.12 cde
	8	637.94 ± 32.95 d	63.54 ± 1.37 d	47.97 ± 0.82 a	6.34 ± 0.36 a
NO_3_^−^	2	413.44 ± 3.40 fg	43.89 ± 0.74 g	29.53 ± 1.03 e	4.67 ± 0.18 de
	4	549.41 ± 5.42 e	49.47 ± 0.35 f	20.24 ± 0.43 g	5.20 ± 0.17 b
	8	425.75 ± 9.84 g	62.39 ± 1.01 d	30.44 ± 1.03 e	5.22 ± 0.14 b
NH_4_^+^	2	941.21 ± 21.69 a	95.16 ± 0.34 a	41.76 ± 1.10 c	5.24 ± 0.27 b
	4	805.66 ± 13.22 b	84.62 ± 0.43 c	46.23 ± 1.43 b	4.93 ± 0.19 bcde
	8	721.17 ± 25.43 c	87.18 ± 2.14 b	33.13 ± 0.44 d	4.98 ± 0.17 bcd

Data in the table are means ± SD, different lowercase letters within a column indicate significant differences between treatments (*p* < 0.05, Duncan).

## Data Availability

All data included in the main text.

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
