# Peer review of "Influence of Nitrogen Supply on Growth, Antioxidant Capacity and Cadmium Absorption of Kenaf (Hibiscus cannabinus L.) Seedlings"

_plants, 2023, doi:10.3390/plants12234067_

Round 1
Reviewer 1 Report
Comments and Suggestions for Authors
Detailed comments are noted on the attached manuscript

Author Response
Please see the attachment。

Reviewer 2 Report
Comments and Suggestions for Authors
The paper „Influence of nitrogen supply on growth, antioxidant capacity, and cadmium absorption by kenaf (Hibiscus cannabinus L.) seedlings” is current and very well structurate.
In the introduction, some concepts from the literature related to the capacity of plants to uptake, transfer and accumulation heavy metals should be introduced.
I propose the publish of the paper after they will expanding the conclusions. In the Conclusion, briefly add the science of your research, benefits, advantages-disadvantages, practical application. Perspectives in the continuation of research and trials.
Reviewer 3 Report
Comments and Suggestions for Authors
The paper focuses on a very relevant issue which is the Influence of nitrogen supply on growth, antioxidant capacity, and cadmium absorption by kenaf (Hibiscus cannabinus L.) seedlings.
The paper deals with important issues:
• 2.1. Phenotypic traits
• 2.2. Chlorophyll content
• 2.3. Uptake, transfer and accumulation of Cd in different parts of kenaf
The authors provided interesting results that showed that adding N fertilizer can promote the growth of kenaf to varying degrees, increase POD, SOD, and CAT enzyme activities, and affect the Cd concentration and accumulation in various parts of kenaf under Cd stress.
The article presents in a legible and transparent manner the material and methods used in a given research work. The methodology is clear and described concisely. Introduction section is comprehensive and is also written in an concise and clear manner. The literature is well-chosen and the conclusions clearly refer to the conducted research.
There are, however some minor weaknesses in discussion, therefore I recommend minor revision of the paper.
Minor issues to be corrected:
• Introduction
It would be good to add a citation to the statement (line 41-42)
It would be good to add a citation to the statement (line 57-58) – “The application of nitrogen (N) fertilizer has become an essential approach to increasing crop yield in China [14]” - too little cited literature to make such an extensive conclusion “
• Results
Table 1. Changes of kenaf morphology in different treatment under Cd stress. - the table is difficult to read - I propose splitting it into two tables to make the results clear.
• Discussion
Reference (line 197-199) - please add the info on the plays an important role in the formation of crop yields
• Discussion
I suggest to extend the literature citation on “Different N forms significantly change chlorophyll content in plants under Cd stress” (line 217) or add literature.
In summary, the paper is worth publishing in the Journal
Reviewer 4 Report
Comments and Suggestions for Authors
The manuscript plants-2696500 lacks novelty because the role of N fertilizers is known. English is bad. The data are wrong in most cases. the parameters are simple to publish in this journal.
Comments on the Quality of English LanguageEnglish very difficult to understand/incomprehensible
Reviewer 5 Report
Comments and Suggestions for Authors
Dear Authors,
The manuscript received for review with the title „Influence of nitrogen supply on growth, antioxidant capacity, and cadmium absorption by kenaf (Hibiscus cannabinus L.) seedlings” is of interest because of the fact that Kenaf (Hibiscus cannabinus L.) has been considered suitable for the remediation of Cd-contaminated farmland, due to its large biomass and resistance to Cd stress. Adding nitrogen N fertilizer is an important measure for increasing crop yields, and may also affect Cd accumulation in plants.
It is recommended to carefully check the way the manuscript was drafted so that it respects the technical editing rules of the journal, as well as the international units of measurement... as well as their correction in the manuscript. But also the detailed presentation of the analytical conditions of the equipment used to determine the monitored parameters.
Congratulations to the authors.

Minor editing of English language required
Reviewer 6 Report
Comments and Suggestions for Authors
General comments
The paper is very interesting, well structured and well written. Some indications are given to improve its presentation.
Specific comments
Table 1.
the font size should be reduced to improve the comprehension of the table.
Figure 2.
The colours should be substituted by different types of patterns. The different types of blue are misleading.
Figure 3.
Idem comment in figure 2
Round 2
Reviewer 4 Report
Comments and Suggestions for Authors
Accept in current form
Author Response
Dear Reviewer,
I would like to express my sincere gratitude for your positive feedback and acceptance of our manuscript. Your prompt response and meticulous review have been a great help to us.
Thank you for your time and effort in improving our manuscript. We are truly grateful for your help.
Best Regards,
Wenlue Li